# Cuban Sugarcane Wax Alcohol Supplementation Prevents Brain and Eye Damages of Zebrafish Exposed to High-Cholesterol and High-Galactose Diet for 30 Weeks: Protection of Myelin, Cornea, and Retina

**DOI:** 10.3390/antiox14121453

**Published:** 2025-12-03

**Authors:** Kyung-Hyun Cho, Ashutosh Bahuguna, Cheolmin Jeon, Sang Hyuk Lee, Yunki Lee, Seung Hee Baek, Chae-Eun Yang, Ji-Eun Kim, Krismala Djayanti

**Affiliations:** Raydel HDL Research Institute, Medical Innovation Complex, Daegu 41061, Republic of Korea

**Keywords:** brain, dyslipidemia, eye, fatty liver, oxidative stress, policosanol, retina, senescence

## Abstract

Cuban sugarcane wax alcohol (policosanol) is a blend of eight characteristic aliphatic alcohols extracted from the Cuban sugarcane and widely recognized for its multifunctional applications and therapeutic properties. In the present study, the potency of policosanol (POL) was assessed for its ability to prevent metabolic stress and associated disorders posed by a high-cholesterol (HC) and high-galactose (HG) diet in zebrafish (*Danio rerio*). Adult zebrafish (*n* = 56/group) were fed either with an HC+HG diet (containing 4%, *w*/*w* cholesterol and 30%, *w*/*w* galactose), or an HC+HG amalgamated diet with POL (final 0.1% *w*/*w* or 0.5% *w*/*w*). Zebrafish in the specified groups were sacrificed post-30 weeks of feeding, and blood and organs (liver, brain, and eyes) were processed for biochemical, histological, and immunohistochemical (IHC) analysis. After 30 weeks of feeding, the highest mortality (12.5%) was noticed in the HC+HG supplement group, which was reduced to 4.5% with co-supplementation of POL (0.1% and 0.5%). In a dose-dependent manner, POL significantly reversed HC+HG elevated levels of total cholesterol (TC), triglycerides (TG), low-density lipoprotein cholesterol (LDL-C), glucose, and malondialdehyde (MDA), while substantially augmenting plasma high-density lipoprotein cholesterol (HDL-C), sulfhydryl content, ferric ion reduction ability (FRA), and paraoxonase (PON) activity. In addition, POL mitigated HC+HG-induced hepatomegaly, inflammation, and fatty liver changes. Consistently, POL minimizes ROS generation and cellular senescence in the brain and substantially improves HC+HG-induced cognitive changes (cessation of swimming ability and motion), with a marked ~5 times higher swimming distance. Notably, POL mitigated the HC+HG-induced corneal opacity and attenuated oxidative stress, apoptosis, 4-hydroxynonenal (4-HNE) accumulation, and myelin sheath degeneration in the retina. The findings underscore the therapeutic potential of policosanol in attenuating oxidative stress, metabolic changes, and various organ damage caused by prolonged exposure to the HC+HG diet.

## 1. Introduction

Excessive galactose consumption in humans, especially for the elderly, triggers the formation of galactitol and non-enzymatic advanced glycation end products (AGEs), which exert detrimental effects on a variety of organs, including the brain and liver [1,2,3]. The accumulation of galactitol negatively affects cellular antioxidants and induces the generation of reactive oxygen species (ROS), thereby damaging cellular macromolecules and compromising cellular membrane integrity [1,4]. Likewise, AGEs have several adverse outcomes, primarily mediated by their interaction with their receptors (RAGEs), which activate nuclear factor kappa B (NF-κB) signaling and stimulate inflammation and oxidative stress [5]. Additionally, AGE/RAGE interactions favored events that affected insulin secretion, β-cell toxicity, and modulated the transcription of genes associated with the progression of type 2 diabetes mellitus [5]. Galactose is associated with premature aging by inducing events like cellular senescence, oxidative stress, and inflammation and displays a high resemblance to the natural aging process [4]. Of note, high-galactose intakes have been recognized for cognitive impairment, neurodegeneration, and hepatic inflammation [6,7]. In addition, a notable effect of galactose has been observed in the induction of ROS generation, protein denaturation, and cellular damage in the eyes, leading to severe complications such as cataracts and blindness [8]. Parallel to galactose, high-cholesterol (HC) [8] and fat consumption [9] also lead to metabolic stress like dyslipidemia and have substantial adverse effects on the liver, kidney, reproductive organs [10,11,12], and coronary heart ailments [13]. The combination of both high-cholesterol and sugar in diet triggers a severe metabolic disturbance that highly resembles the pathophysiology of human obesity, fatty liver, and insulin resistance [14].

Policosanol is a mixture of long-chain aliphatic alcohols (LCAA) of general molecular formula CH_3_-(CH_2_)_n_-CH_3_-OH (n varies from 22 to 32) and can be obtained from diverse plant and insect sources [15]. Policosanol is recognized for its effect on hypertension, ischemic heart disease, intermittent claudication, neurological disorders, and inhibiting platelet aggregation [15,16]. In addition, the effect of policosanol has been recognized for its ability to prevent dyslipidemia and pre-hypertension [15,17,18], particularly its positive impact on the functionality of high-density lipoprotein (HDL) [17,18] and cholesterol efflux capacity [19]. Additionally, policosanol prevents low-density lipoprotein (LDL) from oxidative injury and exhibits a substantial anti-glycation effect [18]. Nevertheless, the composition of policosanol, both in terms of the types and amounts of LCAA, varies considerably depending on the source material’s origin and extraction method; consequently, there is a variation in their functionality [20,21]. For instance, policosanol extracted from Cuban sugarcane wax is a unique blend of eight LCAA (C24, C26, C27, C28, C29, C30, C32, and C34) and has been well recognized for its wide-ranging functionality, which is substantially diverse from the functionality of other policosanols [22]. Especially in Korea, only policosanol sourced from Cuban sugar cane wax is approved by the Ministry of Food and Drug Safety (MFDS) as a functional food to improve dyslipidemia and hypertension [23]. Studies conducted in humans and across various animal models have established the safety and non-toxicity of policosanol. In a post-marketing survey with over 27,000 participants, only 0.07% reported polyurea and weight loss, while 0.05% reported polyphagia following the intake of policosanol [16]. Policosanol consumption at a dose 1500 times higher than the usual human dose showed no evidence of toxicity in animal models, including monkeys, dogs, rats, and mice [16]. Consistently, at 1500 times the usual human dose, policosanol was found safe with respect to reproductive health and fertility in mice and rats [16].

There are studies describing the effect of policosanol on HC diet-induced events [22]; however, a comprehensive investigation of the impact of policosanol on the severity of prolonged dietary consumption of HC and HG remains limited. In view of this, the present study was conducted over 30 weeks to assess the potential effect of policosanol on the HC+HG diet-induced changes in survivability, dyslipidemia, blood, biochemical, and antioxidant profiles in zebrafish. Additionally, histological alterations in the zebrafish liver, brain, and eyes, as well as cognitive impairment, were assessed.

Zebrafish was selected as a model organism due to its high resemblance to the human genome [24]; in particular, important enzymes and receptors involved in zebrafish lipid metabolism are analogous to those in the human lipid metabolism pathway [25]. Also, zebrafish harbor a parallel human brain neurochemistry and highly resemble many behavioral changes observed in humans [26]. Moreover, zebrafish contain nearly all the basic structures of the human eye and thus serve as an ideal model for the ocular disease [27].

## 2. Materials and Methods

### 2.1. Materials

A sample powder of policosanol (POL), extracted from Cuban sugarcane wax, was kindly provided by Raydel Pty. Ltd. (Thornleigh, NSW, Australia). The POL formulation and the certificate of analysis for the used policosanol (batch no. 310030324) is provided in Appendix A. The myelin-specific polyclonal antibodies (0.7 mg/mL) was in-housed produced (in New Zealand Rabbits), which was complementary, obtained from Prof. Cheol-Hee Kim (Laboratory of Developmental Genetics, Department of Biology, Chungnam National University, Daejeon 34134, Republic of Korea). The primary antibodies specific to IL-6 (ab9324) and 4-HNE (ab48056) were procured from Abcam (Cambridge, UK). The fluorescent-tagged antibodies Alexa Fluor^TM^ 405 (A-31556) and Alexa Fluor^TM^ 594 (AB_2534073) were purchased from Invitrogen (Waltham, MA, USA). Unless specified, all additional chemicals and reagents were analytical grade and used as received.

### 2.2. Zebrafish Rearing

A 12-week young wild type AB strain of zebrafish was maintained at a 14 h light/10 h dark photo period in a water tank connected to an automated water supply. Throughout the experimental period, a constant water temperature of 28 °C was maintained, and zebrafish were cultured in accordance with the standard guidelines for Animal Care and Use adopted by Raydel Research Institute (code of approval RRI-23-007, 27 July 2023). Zebrafish were fed with normal Tetrabit (ND, Tetrabit GmbhD49307, Melle, Germany) two times in a day. Before starting the experiment, zebrafish were maintained in the laboratory conditions for one week to acclimate to the environment.

### 2.3. Preparation and Consumption of Different Zebrafish Diets

Normal Tetrabit (ND), a regular fish diet, was blended with 4% (*w*/*w*) cholesterol to prepare the high-cholesterol diet (HC). Similarly, ND was blended with 30% (*w*/*w*) galactose to produce a high-galactose diet (HG). The ND was mixed with 4% (*w*/*w*) cholesterol and 30% (*w*/*w*) galactose to prepare the high-cholesterol + high-galactose (HC+HG) diet. The HC+HG was combined with POL at final concentrations of 0.1% (*w*/*w*) and 0.5% (*w*/*w*) to produce two distinct POL-supplemented diets, named HC+HG+0.1% POL and HC+HG+0.5% POL, respectively.

Zebrafish (*n* = 336) were randomly distributed in six different chambers (*n* = 56/group) and provided with either ND (group I), HC (group II), HG (group III), HC+HG (group IV), HC+HG+0.1% POL (group V), and HC+HG+0.5% POL (group VI). Zebrafish in each group (*n* = 56) were distributed into four tanks (*n* = 14 zebrafish/tank) and fed with their respective diets 10 mg/zebrafish two times per day (~9 a.m. and 6 p.m.), i.e., equivalent to a cumulative 280 mg/tank/day until week 30. Food consumption among the different groups was examined at week 0, week 4, week 10, week 20, and week 30 using the equation: [(cumulative food given per group–residual food)/cumulative food given per group)] × 100. The residual amount of food was determined after 30 min of exposure to the food.

Survival rates were assessed daily, while body weight was measured at baseline (week 0) and subsequently at two-week intervals until week 30.

### 2.4. Corneal Clouding and Severity Index

At 30 weeks of feeding, eyes were analyzed using a Motic SMZ 168 stereomicroscope (Hong Kong, China) connected with a digital camera (Motic cam 2300, Hong Kong, China). Prior to the eye examination, zebrafish were anesthetized by submerging them in a 2-phenoxyethanol solution (0.1%). The eyes of anesthetized zebrafish were immediately visualized under the stereomicroscope to observe the corneal clouding. Subsequently, images were captured and processed for the semi-quantitative assessment of the cloud severity index (SI) using a circular Cho and Ashu scale (designed and developed by Cho and Ashu, Appendix A). The severity was scored using the classification [grade 0: no clouding; grade 1: very small clouding (less than 1/8 area of lens); grade 2: small clouding (area between 1/8 and 1/4 of lens); grade 3: medium clouding (area between 1/4 and 1/2 of lens); grade 4: upper-medium clouding (area between 1/2 and 3/4 of lens); and grade 5: high clouding (area > 3/4 of lens). The SI for each group was calculated using the following equation:$$Severity\;index\;(SI)=\frac{\sum \left[Grade\times number\;of\;eyes\;in\;that\;stage\right]}{Total\;number\;of\;eyes}$$

### 2.5. Blood and Organs Collection

Following 30 weeks of dietary intervention, zebrafish were euthanized via hypothermic shock by placing them in ice sludge. Gill movements were carefully analyzed as a marker of death. Promptly, blood from 6 zebrafish/tank of the specified group was collected and pooled (~25 μL), followed by mixing with 1 mM ethylenediaminetetraacetic acid (EDTA)-phosphate-buffered saline (PBS) in 2:3 (*v*/*v*). Prior to blood collection, zebrafish from all groups were deprived of food overnight, and blood was collected from all groups between 9 a.m. and 10:30 a.m. to minimize circadian and feeding-associated differences. Blood from the different groups was centrifuged at 6000 rpm for 10 min to separate blood cells and plasma. The supernatant (plasma) was collected and stored in a refrigerator (4 °C) for further analysis.

Organs (liver, brain, and eyes) were surgically excised from the individual zebrafish, weighed, and preserved in 10% formalin for histological examination.

### 2.6. Blood Biochemical Examination

Blood total cholesterol (TC), triglycerides (TG), high-density lipoprotein cholesterol (HDL-C), and plasma liver function enzymes aspartate aminotransferase (AST) and alanine aminotransferase (ALT) were quantified utilizing the commercial kits and adopting the methodology recommended by the manufacturers. A detailed procedure is outlined in Appendix A.

The glucose level of blood was determined using the digital blood glucose meter (AccuCheck, Roche, Basel, Switzerland) whereas the malondialdehyde (MDA) and ferric ion reduction ability (FRA) of plasma was quantified by the earlier descried method [28]. A detained procedure of MDA and FRA determination is explained in the Appendix A.

The plasma sulfhydryl content was quantified using 5,5′-dithiobis-(2-nitrobenzoic acid) (DTNB) method [28]. In brief, 50 μL of DTNB (4 mg/mL) was mixed with 50 μL of plasma (1 mg/mL protein). Following 2 h incubation at room temperature (RT), absorbance 412 nm was recorded, and sulfhydryl content was expressed as mmol/mg protein, quantified by using the molar absorbance coefficient (ε) 1.36 × 10^4^ M^−1^cm^−1^ of the 5-thiol-2-nitrobenzoic acid (formed product).

To determine the paraoxonase (PON) activity [28], 40 μL of plasma (1 mg/mL protein) was mixed with 160 μL of 0.15 g/mL paraoxon ethyl. Following 2 h incubation at RT absorbance 415 nm was determined and PON activity was expressed as μU/L/min using molar absorbance coefficient (ε) 1.7 × 10^3^ M^−1^cm^−1^ of *p*-nitrophenol (formed product).

### 2.7. Zebrafish Embryo Collection and Microinjection of Plasma

Young male and female zebrafish (18 weeks old) were placed in the breeding tank and kept separated overnight by a physical divider. In the morning (~8.30 a.m.), the divider was ejected, and the male and female zebrafish were allowed to pair uninterruptedly (approximately 30 min) in the dark. The produced embryos were retrieved and maintained in a sea salt solution (0.012% *w*/*v* comprising 1 μg/mL methylene blue). Embryos at the 16-cell stage (1.5 h post-fertilization) were assigned to 6 distinct groups (*n* = 150/group) and microinjected with the plasma following the earlier described method [28]. Embryos in group I received a 10 nL microinjection of the plasma obtained from the ND-supplemented group, whereas embryos in groups II, III, and IV received the 10 nL microinjection of the plasma obtained from the HC-, HG-, and HC+HG-supplemented group. A 10 nL plasma obtained from the 0.1%POL+HC+HG and 0.5%POL+HC+HG group was injected into the embryos of group V and group VI, respectively. The plasma (10 nL) was injected under a microscope using a glass capillary needle mounted on a microcapillary pipette (PC-10; Narishige, Tokyo, Japan), connected to a pneumatic picopump (PV830; World Precision Instruments, Sarasota, FL, USA) equipped with a magnetic manipulator (MM33; Kantec, Bensenville, IL, USA). The nearly same position of the yolk in embryos was used as a site for microinjection to minimize bias.

Embryo viability and developmental changes were examined by visualizing the embryos under a stereomicroscope (Motic SMZ 168; Hong Kong, China) equipped with a Motic cam 2300 CCD camera. Embryos were examined under a microscope during 72 h post-injection in accordance with the OECD 2019 guidelines [29]. The swimming abilities of zebrafish across the group were examined at 144 h post-treatment. For the swimming activity evaluation, eight embryos (*n* = 8/group) were placed in a 48-well culture plate containing 300 μL of sea salt solution. After 10 min acclimatization, embryos were visualized under the microscope for 1 min, and a video was recorded (for 30 s swimming activity).

### 2.8. Dihydroethidium (DHE) and Acridine Orange (AO) Fluorescent Staining of Embryos

DHE and AO fluorescent staining was performed following the method described earlier [30]. In brief, embryos (*n* = 10) were kept in a 48-well culture plate containing 250 μL solution of a DHE (30 μM) and AO (5 μg/mL). After 30 min incubation in the dark, stained samples were washed with PBS (three times) and examined under a Nikon EclipseTE2000 fluorescent microscope (Tokyo, Japan) at 585 nm/615 nm of excitation and emission wavelengths for DHE and 505 nm/535 nm for AO.

### 2.9. Histological Evaluation, Cellular Senescence, and Fluorescent Staining

Different organs were individually embedded in FSC22 frozen solution (Leica, Nussloch, Germany) to develop tissue-embedded solid blocks, which were then processed for cryo-sectioning (7 μm thick) using a Leica cryo-microtome (CM-1510S, Nussloch, Germany). For the evaluation of histological changes in the liver, brain, and eyes, the respective sections (7 μm thick) were stained with hematoxylin and eosin (H&E), adhering to the previously established protocols [31].

Oil red O (ORO) staining of the liver section was performed by the earlier described method [30]. Briefly, the liver section (7 μm thick) was immersed in 500 μL of 0.1% ORO solution and incubated for 5 min at 60 °C. The stained section was washed with 60% isopropanol (three times) and visualized under the microscope.

Cellular senescence in the tissue section was detected by the senescence-associated β-galactosidase staining [32]. Briefly, the tissue section (7 μm thick) was flooded (~750 μL) with 5-bromo-4-choloro-3-indolyl-β-D-galactopyranoside solution (0.1%). After 18 h incubation at RT (in a moist environment), the section was rinsed with PBS (three times) and examined microscopically to detect blue-stained senescent-positive cells.

The fluorescent staining for DHE and AO in the different tissue sections was performed using the methodology described in Section 2.8.

### 2.10. Immunohistochemical (IHC) Staining

For the detection of interleukin (IL)-6, the liver section (7 μm thick) was covered with IL-6-specific primary antibodies (200× diluted, Abcam ab9324) for 16 h in a cold (4 °C) and moist environment. Subsequently, the section was developed using the EnVision HRP polymer kit (Dako, Glostrup, Denmark), containing anti-IL-6 secondary antibody tagged with HRP. The developed IHC section was visualized under a microscope, and the images were transformed into red conversions to enhance the visibility employing the ImageJ software (https://imagej.net/ij, version 1.53, assessed on 6 June 2025). The red conversion was conducted at the brown-color threshold value 20–120. The myelin sheath in the eye retina was detected by exposing the eye section (7 μm thick) with myelin-specific polyclonal antibodies from a rabbit (200× diluted, in house produced by Prof. Cheol-Hee Kim, Laboratory of Developmental Genetics, Department of Biology, Chungnam National University, Daejeon 31134, Republic of Korea). After 16 h incubation in a humid and cool environment (4 °C), the section was washed three times with PBS and subsequently treated with 400× diluted fluorescent-tagged goat anti-rabbit IgG (AlexaFluor^TM^ 405, A-31556, Invitrogen, Waltham, MA, USA) for 4 h at 4 °C. For visualization, the section was rinsed with PBS and then examined using a fluorescent microscope at excitation and emission wavelengths of 401 nm and 422 nm, respectively.

Accumulation of 4-hydroxynonenal (4-HNE) in the brain and eye retina was determined by the IHC staining. The brain and eye sections (7 μm thick) were incubated with the 200× diluted 4-HNE specific antibodies (Abcam ab48056, Cambridge, UK) for 16 h in a moist, cool (4 °C) environment. Thereafter, the bond primary antibodies were detected using the 400× diluted fluorescent tagged secondary antibody (Alexa Fluor^TM^ 594, AB_2534073 Invitrogen, Waltham, MA, USA). The stained section was imaged using a fluorescent microscope with the excitation wavelength set to 590 nm and the emission wavelength set to 618 nm.

### 2.11. Behavioral Analysis

Behavioral analysis of zebrafish post-30-week consumption of the designated diets was examined in an open-field tank (25 cm × 6 cm × 20 cm) equipped with a video recording facility. The tank was divided into two equal sectors by a horizontal line and named as the lower and top sectors (as depicted in Appendix A). Zebrafish (43 weeks old) from various groups were individually placed in a divided tank and allowed to acclimate for 30 min to the environment. After 30 min of acclimatization, the swimming activity of the zebrafish was monitored for 1 min, and the swimming pattern (1 min) was recorded using a digital camera (Canon EOS 90D, Tokyo, Japan). The recorded video was processed using Any-Maze software (version 7.0, Kim and friends, Seoul, Republic of Korea) to analyze the swimming trajectories, total swimming distance, and swimming time in the top half. Notably, for each group, four zebrafish (*n* = 4) were analyzed for swimming activity, and the quantitative results are depicted as mean ± SD across four independent experiments.

### 2.12. Stastistical Analysis

The SPSS software (version 29, Chicago, IL, USA) was used to determine statistical differences. After confirming the data’s normality, the statistical difference between the groups was analyzed using one-way analysis of variance (ANOVA) at a 95% confidence level. Post hoc comparisons were performed using Tukey’s analysis with statistical significance defined as *p* < 0.05.

## 3. Results

### 3.1. Evaluation of Survivability, Body Weight Change, and Morphometric Parameter

The zebrafish survivability among the groups varied substantially during 30 weeks of feeding (Figure 1A). Until week 7, 100% zebrafish survivability was observed among the groups. The first decline in survivability (96.4%) was reported in week 8 in the HC-supplemented group, which further declined to 92.8% in week 14 and finally reached 89.3% in week 30 (Figure 1A). The most deleterious effect on zebrafish survivability was observed in the HC+HG consumption group, where 95.9% survivability was reported in week 10, which progressively declined with time and finally reached 87.5% in week 30. In contrast, in the HG-supplemented group, 100% survivability was observed up to week 17, which declined slightly to 95.3% in week 30. The maximum zebrafish survivability of 96.9% was observed in the ND, 0.1% and 0.5% POL-supplemented groups at week 30.

The body weight analysis revealed a significant increase over time across all groups (Figure 1B and Appendix A). Compared to the body weight on the initial day (week 0, 307.7 mg), the least gain (64.5%) in body weight was observed in the ND-supplemented group after 30 weeks (506.1 mg). In contrast, higher weight gains of 86.1% and 72.8% were observed in the HC- and HG-supplemented groups, respectively. After 30 weeks of supplementation, the most notable body weight gain of 113.6% (657.3 mg) was noticed in the HC+HG-supplemented group compared to the week 0 body weight (307.6 mg), which accounted for a significant 29.9%, 13.2% and 18.5% higher than the body weight of the ND-, HC-, and HG-supplemented groups at week 30. A co-supplementation of 0.1% and 0.5% POL substantially inhibited the HC+HG-triggered body weight gain, indicated by a significantly 16.5% and 18.0% lower body weight than the body weight of HC+HG-consuming zebrafish at week 30. Compared to week 0, body weight gains of 77.4% and 74.1% were observed in the 0.1% and 0.5% POL-supplemented groups, respectively, against a 113.6% weight gain in the HC+HG-supplemented group.

The morphometric analysis showed an ~11% reduction in the body length (BL) to body depth (BD) ratio in the HC+HG consumption group, compared to the ND- and POL-supplemented groups; nevertheless, this variation was not statistically significant (*p* > 0.05) (Figure 1C,D). A uniform food consumption efficacy (~95–100%) that remains consistent at different time points (week 0, week 4, week 10, week 20, and week 30) was observed among all the groups, highlighting that the different dietary formulations did not have any impact on the zebrafish appetite and liking or disliking towards the food.

### 3.2. Organ Morphology and Organ Weight

The organ morphology and weight, as depicted in Figure 2, indicated an adverse effect of HC+HG consumption on liver and brain morphology and weight. The 30-week consumption of HC+HG resulted in hepatomegaly with a significant 3.8 times elevated liver weight relative to the ND-supplemented groups (Figure 2A,B). The co-supplementation of POL at both tested concentrations (0.1% and 0.5%) effectively prevented HC+HG-induced hepatomegaly, with a significant ~3 times reduction in liver weight. The normal size and weight of the brain, as observed in the ND (control) group, were significantly compromised by HC+HG consumption, as evidenced by the 1.6 times decrease in brain weight in the HC+HG-supplemented group compared to the ND group. The co-supplementation of POL (at 0.1% and 0.5%) effectively prevented the HC+HG alteration of brain morphology and significantly elevated the brain weight by ~1.5 times (*p* < 0.001) (Figure 2A,C). Interestingly, POL supplementation resulted in liver and brain weights that were statistically similar to those noticed in the ND (control) group, highlighting the impact of POL in preventing HC+HG-triggered changes in organ morphology and weight to basal levels.

### 3.3. Blood Lipid Profile and Glucose Level

The utmost TC (342.1 ± 23.5 mg/dL), TG (227.8 ± 15.6 mg/dL), and LDL-C (271.2 ± 23.2 mg/dL) levels were noticed in the HC+HG consumption group which were substantially higher, by 2.7 times (125.6 ± 11.8 mg/dL), 2.2 times (104.3 ± 18.2 mg/dL), and 8.1 times (33.2 ± 17.9 mg/dL), than the baseline values observed in the ND control group (Figure 3A–C). Compared to the individual supplementation of HC and HG, a joint supplementation (i.e., HC+HG) exhibited significantly increased plasma levels, ~1.2 times higher, of TC, TG, and LDL-C. The HC+HG-disturbed plasma lipoprotein levels were substantially mitigated by the consumption of POL in a dose dependent manner, manifested by a 1.5 times (235.9 ± 26.1 mg/dL) and 1.9 times (182.1 ± 26.4 mg/dL) reduction of TC, a 1.4 times (167.5 ± 14.9 mg/dL) and 1.6 times (144.5 ± 19.5 mg/dL) reduction of TG, and a 1.7 times (158.7 ± 26.8 mg/dL) and 2.8 times (98.5 ± 25.1 mg/dL) reduction of LDL-C levels in the 0.1% and 0.5% POL-supplemented groups, respectively, compared to the HC+HG-supplemented group. The lowest HDL-C level (25.4 ± 5.9 mg/dL), i.e., substantially 2.8 times lower than the HDL-C level (71.5 ± 7.5 mg/dL) in the ND (control) group, was found in the HC+HG group (Figure 3D). The HC+HG-diminished HDL-C levels were significantly elevated by 1.7 times (43.7 ± 8.1 mg/dL) and 2.2 times (54.7 ± 7.5 mg/dL) by the co-supplementation of 0.1% and 0.5% POL, respectively. Consistently, the HC+HG-elevated TG/HDL-C ratio (9.6) significantly decreased to 3.9 and 2.6 by the intake of 0.1% and 0.5% POL, respectively (Figure 3E).

The highest blood glucose level (94.3 ± 8.4 mg/dL) was observed in the HC+HG-supplemented group, which was statistically similar to the glucose level detected in the HC- and HG-supplemented groups (Figure 3F). The co-supplementation of 0.1% POL (73.8 ± 4.9 mg/dL) and 0.5% POL (64.6 ± 7.7 mg/dL) effectively countered the HC+HG elevated blood glucose level (94.3 ± 8.4 mg/dL) in a dose-dependent manner.

### 3.4. Plasma Levels of Oxidative Species, Antioxidant Variables, and Hepatic Function Biomarkers

The maximum MDA level (14.4 ± 0.5 μM) was detected in the HC+HG group, which was notably 2.7 times more than the MDA level (5.6 ± 0.6 μM) detected in the ND (control) group (Figure 4A). The supplementation of POL at 0.1% and 0.5% reduced the HC+HG elevated MDA level by 1.5 times (9.8 ± 0.7 μM) and 1.8 times (8.2 ± 0.7 μM), respectively. The basal level of the plasma sulfhydryl content (13.5 ± 0.4 nmol/mg), ferrous equivalent (481.5 ± 20.1 μM), and PON activity (10.9 ± 1.1 μM) was significantly diminished by 1.6 times (8.4 ± 0.5 nmol/mg), 1.8 times (267.4 ± 28.2 μM), and 3 times (3.6 ± 0.8 μM) by the intake of HC+HG which was elevated considerably by 1.3~1.5 times (11.1 ± 0.3 and 12.3 ± 0.2 nmol/mg), 1.3~1.5 times (359.5 ± 17.4 and 400.1 ± 29.1 μM), and 1.8~2.2 times (6.5 ± 1.2 and 7.9 ± 0.9 μM) by the supplementation of 0.1% and 0.5% POL, respectively (Figure 4B–D). When compared with the 0.1% POL-supplemented group, the 0.5% POL-supplemented group exhibited significantly (*p* < 0.01) 10.8%, 11.3%, and 21.5% higher plasma sulfhydryl content, FRA, and PON activity, respectively, establishing the dose-dependent impact of POL in managing the HC+HG-triggered adverse events in plasma.

The HC+HG elevated plasma hepatic function biomarkers AST (764.5 ± 58.9 IU/L) and ALT (737.9 ± 81.9 IU/L) levels were significantly reduced to 622.4 ± 42.1 IU/L and 555.7 ± 23.5 IU/L, and 615.8 ± 34.8 IU/L and 547.4 ± 30.4 IU/L, respectively, by the co-supplementation of 0.1% and 0.5% POL (Figure 4E,F). Nevertheless, the 0.5% POL concentration was most effective, lowering the HC+HG elevated AST and ALT levels by 10.7% and 11.1% more, respectively, than the 0.1% POL-supplemented group.

### 3.5. Effect of Plasma Injection on the Zebrafish Embryos

The plasma functionality obtained from the different groups was assessed by injecting it into zebrafish embryos. The survivability of embryos varied significantly in response to the injection of plasma from different groups (Figure 5A). The maximum 82% survivability at 72 h post-injection was observed in the embryos that received microinjections of plasma from the ND group. In contrast, the lowest embryo survivability (50%) at 72 h post-injection was observed in embryos that received microinjections from the plasma of the HC+HG group, which was statistically similar (*p* > 0.05) to the embryo survivability observed in the plasma injected from the HC- and HG-supplemented groups. The plasma obtained from the 0.1% and 0.5% POL-supplemented groups displayed 66% and 69% embryo survivability at 72 h post-injection, which accounts for a significantly 32% and 38% higher survivability than that observed in the HC+HG group.

The morphological examination (72 h post-injection) revealed that many embryos that received plasma from the HC+HG groups emerged with developmental abnormalities, primarily limited to tail fin curvature (indicated by blue arrowhead), yolk sac edema (indicated by red arrowhead), and diminished somite counts (Figure 5B,C,F). Likewise, many embryos in the HC and HG groups also showed developmental defects and reduced somite number. Compared with the HC+HG group, embryos receiving plasma from the POL group showed minimal developmental defects and higher somite counts (~30), comparable to the average somite counts of the ND group (~31).

The intensified red and green, fluorescent intensities corresponding to dihydroethidium (DHE) and acridine orange (AO) staining revealed the highest ROS production and apoptosis in the embryos from the HC+HG group, which was notably 3.2 times and 2.6 times higher than the respective intensities observed in the ND-supplemented group (Figure 5D,E,G). Compared to the HC+HG group, a significant 2.4 times and 2.8 times reduction in DHE fluorescent intensity, and 1.9 times and 3.1 times reduction in AO fluorescent intensity appeared in the 0.1% and 0.5% POL-supplemented groups, respectively. No significant changes in DHE and AO fluorescent intensities were observed in the HC and HG groups with respect to the HC+HG group.

### 3.6. Swimming Behavior of Zebrafish Embryos

At 144 h post-injection, approximately 30% of the surviving embryos in the HC+HG group exhibited a severe teratogenic defect marked by tail fin curvature (indicated by blue arrowhead), bending of the back (indicated by green arrowhead), and yolk sac edema (indicated by red arrowhead) and pericardial edema (indicated by brown arrowhead) (Figure 6). Notably, most of these embryos emerged with an uninflated swimming bladder, with altered stationary posture and severely compromised swimming activity (Figure 6 and Appendix A). Contrary to this, the majority of embryos in the ND, 0.1% and 0.5% POL groups displayed an inflated swimming bladder (indicated by pink arrow), proper stationary posture, and swimming activity with no severe developmental defects.

### 3.7. Liver Histology and Immunohistochemical Analysis

The liver H&E staining revealed the heightened neutrophil infiltration in the HC+HG group, which was markedly 2.5 times and 1.5 times higher than the neutrophil counts in the HC and HG groups and 5.4 times more than the neutrophil numbers observed in the ND (control) group (Figure 7A,B,E). The co-supplementation of POL at both 0.1% and 0.5% significantly reduced the HC+HG-provoked neutrophil infiltration by 3.2 times and 4.7 times, respectively.

Like the H&E staining, the IHC staining revealed a massive 7.3 times higher interleukin (IL)-6 production in the hepatic section of the HC+HG-supplemented group compared to the basal IL-6 level detected in the ND (control) group (Figure 7C,D,F). The HC+HG-induced IL-6 levels were effectively mitigated by POL supplementation in a dose-dependent manner, as reflected by a significant 2.4 times and 5.5 times decrease in IL-6 production in the 0.1% and 0.5% POL-supplemented groups, respectively, compared to the HC+HG group. Compared with the 0.1% POL group, the 0.5% POL-supplemented group showed a 2 times greater efficacy in reducing the HC+HG-provoked IL-6 level.

### 3.8. Liver Steatosis, Reactive Oxygen Species, and Senescence

The ORO staining exhibited a heavy lipid aggregation (steatosis) in the liver of the HC+HG group that was quantified 14.6 times higher compared to the ORO-stained area spotted in the ND (control) group (Figure 8A,D). POL supplementation showed an inhibitory effect on the HC+HG-induced steatosis, as reflected by a notable 4.5 times and 10.1 times decline in ORO-stained areas in the 0.1% and 0.5% POL-supplemented groups compared to the HC+HG group.

The DHE and SA-β-gal staining indicated a basal level of ROS and cellular senescence in the ND (control) group which was significantly augmented by 8.7 times and 3.3 times, respectively, following supplementation with HC+HG (Figure 8B,C,E,F). Supplementation with 0.1% and 0.5% POL resulted in a notable 3.5- and 7.1-fold reduction in ROS production, as well as 1.5- and 1.8-fold reductions in cellular senescence, compared with the HC+HG group, highlighting the inhibitory effect of POL on HC+HG-induced adverse effects.

### 3.9. Brain Histology

The H&E staining documented non-substantial morphological changes in the brain across the groups. However, compared to the ND and POL groups, a notably high presence of vacuolation and mononuclear cells with a clear zone had appeared in the tectum optic (TeO) and periventricular gray zone (PGZ) of the brain from the HC+HG, HC, and HG supplement groups (Figure 9A,B, and Appendix A).

The DHE and AO fluorescent staining spotted a heightened ROS generation and apoptosis in the HC+HG-supplemented group, mainly concentrated on the PGZ of TeO near to torus longitudinalis (TL) and lateral division of valvular cerebelli (Val) (Figure 9C–E,I,J). The HC+HG-induced ROS generation was substantially minimized by 1.7 times and 2.1 times, and apoptosis by ~2 times following the supplementation of 0.1% and 0.5% POL, respectively.

The IHC staining revealed a significantly 26.2 times higher accumulation of 4-hydroxynonenal (4-HNE) around the vascular lacuna of the area postrema (Vas) in the HC+HG-supplemented group than in the ND group, which was significantly 2.5 times and 3.4 times inhibited by the co-supplementation of 0.1% and 0.5% POL (Figure 9F,K). Consistent with the IHC staining, the SA-β-gal staining documented the highest presence of senescent positive cells (~9%) in the brain of the HC+HG supplement group, which was significantly reduced by up to 4.1% and 4.5% following the co-supplementation of 0.1% and 0.5% POL (Figure 9G,H,L).

### 3.10. Zebrafish Swimming Behavior

A variable swimming pattern was observed in the zebrafish across the groups (Figure 10A,B, and Appendix A). The ND group displayed a typical swimming trajectory (moving freely across the tank), with an average swimming distance of 3.9 m and a 16 s stay time in the top half of the tank (Figure 10A–D). In contrast, zebrafish swimming activity was significantly hampered in the HC- and HG-supplemented groups. In the HC group, zebrafish movement is highly restricted around the central part of the tank, with a swimming distance of 2.3 m and a 5.3 s stay time in the top half. Zebrafish in the HG group showed thigmotaxic behavior (movement along the tank wall) that was highly restricted to the lower half of the tank, with an average swimming distance of 2.4 m. The most detrimental impact on swimming activity was noticed in the zebrafish of the HC+HG group, where severe cessation of swimming activity was noted, with the fish confined to the bottom of the tank and a significantly reduced average swimming distance of 9.7 times compared to the ND groups (3.9 m). The HC+HG-impaired swimming activity was substantially recovered in a concentration-dependent manner by the co-supplementation of POL. The swimming trajectories for the 0.1% and 0.5% POL-supplemented groups depict the free movement of zebrafish in the top and lower halves of the tank, with significant 5.6- and 6.8-fold increases in swimming distance over the HC+HG group. However, compared with the 0.1% POL-supplemented group, the 0.5% POL-supplemented group showed a more protective effect, characterized by a 15.3% increase in swimming distance and a 23.5% increase in stay time in the top half.

### 3.11. Corneal Opacity in Zebrafish Eyes

Post-30 weeks intake of HC+HG diet, 97.3% corneal clouding was noticed in the eyes of the zebrafish (Figure 11A,B). Also, 59.7% and 74.9% of zebrafish in the HC- and HG-supplemented groups emerged with corneal clouding. Contrary to this, only 65.8% and 52.7% incidences of corneal clouding were noticed in the 0.1% and 0.5% POL group, which were significantly 1.5 times and 1.9 times lower than the incidence of corneal clouding that appeared in the HC+HG-supplemented group. Consistent with the high incidence of corneal clouding, the highest clouding severity index (2.8) was observed in the HC+HG group, which was significantly reduced by 57.6% and 68.8% with the co-supplementation of 0.1% and 0.5% POL, respectively.

In addition to the severe corneal opacity, a notable change in the iris color had appeared in the HC+HG supplement zebrafish (Figure 11A,C). A characteristic silver bluish-green color of the iris, as appeared in the ND (control) group, transformed into a pale silver-gray color in the HC+HG-supplemented group. In contrast, a characteristic silver bluish-green iris color was observed in the POL-supplemented groups, indicating a counter-effect of POL supplementation on the HC+HG transformed iris color.

### 3.12. Histology of Eyes (Retina)

The H&E section of the eyes (retina) showed a substantial histological variation among the groups (Figure 12A). Unlike the ND group, high vacuolation was observed in the ganglion cell layer (GCL) of the retina in the HC-, HG-, and HC+HG-supplemented groups. Similarly, compared to the ND group, reduced nuclei density in the inner nuclear layer (INL) was noticed in the retina of the HC-, HG-, and HC+HG-supplemented groups. The most notable changes concerning the outer nuclear layer (ONL) disorganization were seen in the HC+HG supplement group. The co-supplementation of POL at both tested amounts (0.1% and 0.5%) prevented the histological changes in the retina induced by HC+HG, as reflected by reduced vacuolation in the GCL, densely packed nuclei in the INL, and a well-organized ONL, compared to the HC+HG-supplemented group.

The IHC staining targeting myelin shows the highest fluorescent intensity in the ND-supplemented group (Figure 12B,G). In contrast, the myelin-specific fluorescent intensity was severely compromised in the HC+HG-supplemented group, resulting in a significant 7.9 times reduction compared to the ND (control) group. The HC+HG-triggered demyelination was substantially prevented by the co-supplementation of POL, reflected by 4.8 times and 5.9 times higher fluorescent intensity in the 0.1% and 0.5% POL-supplemented groups, respectively, relative to the fluorescent intensity of the HC+HG group.

The HC+HG group exhibited extensive ROS production and apoptosis, as visualized by DHE and AO staining, which was significantly reduced by 1.8~2.4-fold and 1.9~2.5-fold by the co-supplementation of 0.1% and 0.5% POL (Figure 12C–E,H,I). However, compared to the 0.1% POL, a significantly ~25% lower ROS and apoptosis were noticed in the 0.5% POL-supplemented group.

The IHC staining demonstrated pronounced accumulation of 4-HNE around the retinal pigmented epithelium (RPE) and the photoreceptor layer (PRL), which was significantly 2.3 times and 2.8 times reduced by the supplementation of 0.1% and 0.5% POL (Figure 12F,J). The combined results of eye histology demonstrated the effectiveness of POL in preventing HC+HG-induced ocular damage.

## 4. Discussion

Consuming high amounts of cholesterol has significant deleterious effects [9,10,11,12,13,14,15,16,17,18,19,20,21,22], primarily by inducing oxidative stress and impairing antioxidant defenses [33]. Likewise, high sugar consumption is associated with the induction of oxidative stress [34], inflammation [35], and injurious effects on various organs [34]. In combination, HC+HG served as an ideal dietary model for the indication of obesity and metabolic disorders mimicking the pathophysiology and disease in humans [14]. Consistently, we observed a significant body weight gain post-30 weeks of intake of the HC+HG diet compared to the body weight at week 0, which was effectively protected by the consumption of POL at 0.1% and 0.5%. The results are supported by earlier findings that document the anti-obesity role of POL [15]. As a key mechanism, POL augments energy expenditure in adipose tissue and enhances brown adipose tissue activity [15,36], thus impact on diet-induced obesity [37]. Even in human subjects, POL displayed an anti-obesity effect, characterized by a notable reduction in body weight after 12 weeks of POL intake [17].

In addition to its anti-obesity effect, POL effectively countered the HC+HG-induced dyslipidemia. Results are supported by earlier studies that document the positive impact of POL in managing dyslipidemia, primarily by targeting 3-hydroxy-3-methylglutaryl-coenzyme A (HMG-CoA) reductase, an essential rate-limiting enzyme in cholesterol biosynthesis [15,36]. In addition, a positive modulatory effect of hexacosanol (an important LCAA of POL) on the nuclear transition of the sterol regulatory element-binding protein (SREBP)-2 has been described, which ultimately impacts the expression of HMG-CoA reductase [38]. In addition, the substantial effect of POL on cholesterol efflux capacity [19] and cholesterol catabolism in the liver [39], which enables the transformation of cholesterol into bile acid and further fecal excretion [39], has been described as an important event contributing to the cholesterol-lowering effect. As with total cholesterol, a lower LDL-C level was observed in the POL-supplemented groups, suggesting a positive effect of POL on the reduction in HC+HG-induced LDL-C levels. The findings corroborate earlier reports, which represent the inhibitory impact of POL on apolipoprotein B (apo-B) expression (a vital protein component of LDL-C) and the upregulation of the LDL receptor (LDL-R) in the liver, prompting the clearance of LDL-C from the blood [39]. A noteworthy impact of a 30-week POL intake on the augmentation of HC+HG-diminished HDL-C was observed. The effect of POL on the augmentation of HDL-C is related to previous reports suggesting POL’s inhibitory role towards cholesteryl ester transfer protein (CETP) [18] and upregulation of apolipoprotein A-I (apoA-I, a major HDL protein) [19].

The adverse effects of a high-fat diet, galactose, and oxidative stress have been recognized in relation to insulin secretion and insulin sensitization, which can lead to hyperglycemia [40,41]. Herein, we also noticed that the co-supplementation of POL substantially prevented the hyperglycemia induced by the consumption of HC+HG. The accumulating literature suggests that POL has an affirmative effect on insulin sensitization and secretion, mediated by the AMPK/PI3K/Akt signaling pathways, leading to hypoglycemia [42]. High cholesterol and galactose intake emerged with severe oxidative stress reflected by elevated plasma MDA levels and diminished sulfhydryl groups, which are important stress markers correlated with various ailments [43,44,45]. In response to POL, substantial reductions in HC+HG-induced MDA levels and elevations in sulfhydryl groups, as well as increased FRA and PON activities, were observed, attesting to the effectiveness of POL against HC+HG-induced oxidative stress and disturbed antioxidant parameters. The findings are in good agreement with earlier studies documenting the beneficial impact of POL on plasma MDA [46], FRA, and PON activities [32] altered by various external stresses. Also, POL has been recognized to activate AMPK by phosphorylating threonine (Thr172) [47], which plays a critical role in the ROS defense system by inducing the expression of antioxidants such as superoxide dismutase and catalase [48]. In addition, the antiglycation effect of POL has been described [18], which is also helpful in minimizing oxidative stress, as high sugar consumption initiates non-enzymatic glycation, leading to the formation of advanced glycation end products (AGEs) that induce oxidative stress, inflammation, and disturbed redox homeostasis [49]. Furthermore, the experiment carried out on zebrafish embryos validated poor plasma health in the HC+HG-supplemented group, which substantially improved following the co-supplementation of POL, as evidenced by markedly higher survivability, elevated somite counts, and normal swimming behavior, analogous to the ND (control) group. The lower MDA level and higher antioxidant variables in plasma consumed with POL are the reasons for the improved embryo survivability and developmental effects. However, a detailed mechanistic study is needed to reach a confirmatory conclusion.

HC has been documented to induce fatty liver disease [50], while high-galactose consumption is associated with severe oxidative stress, senescence, and efflux of pro-inflammatory cytokines, such as TNF-α, IL-6, and IL-10, in the liver [51]. Consistent with this, higher lipid accumulation, neutrophil infiltration, cellular senescence, IL-6, and ROS production were noticed in the HC+HG-supplemented group, which was substantially prevented by the co-supplementation with POL, highlighting the hepatoprotective nature of POL. The outcomes are consistent with earlier studies reporting the hepatoprotective effect of POL against various external stresses [46,52]. The counter-effect of POL on dyslipidemia and hyperglycemia is one of the major reasons behind the improved hepatic health in the POL-supplemented group. This notion is in accordance with the existing literature highlighting the adverse effect of dyslipidemia [53] and hyperglycemia [54] on the liver. In addition, higher ROS production was noticed in the HC+HG group, which was effectively curtailed by POL supplementation, leading to hepatoprotection, as oxidative stress has been recognized as a major contributor to the development of fatty liver [55] and hepatic damage [56]. In addition, the lowest cellular senescence was detected in the POL-supplemented group, the least ROS production in response to POL being the driving factor for the lowered senescence, as a direct relation between oxidative stress and senescence has been described [57].

High galactose consumption has been recognized for its role in generating ROS and augmenting oxidative stress, which provokes cognitive changes and brain aging [58,59]. A similar effect was observed in the present study, where a high ROS generation, severe alteration in the swimming behavior (cognitive change), and brain aging (reflected by cellular senescence) were noticed in response to HC+HG consumption, which was significantly mitigated by the co-supplementation of POL. The positive effect of POL on inhibiting ROS generation in the brain is considered a key event in preventing brain damage induced by HC+HG, as ROS contributes substantially to cognitive alterations and brain aging [60]. A massive buildup of lipid peroxidation product 4-HNE was noted in the brain of HC+HG groups, which has been well established to induce oxidative stress and apoptosis [61]. An inhibitory effect of POL on the accumulation of 4-HNE in the brain suppresses apoptosis and protects the brain from the detrimental effects of HC+HG. In addition, high-galactose catalyzed non-enzymatic glycation leads to the formation of AGEs [58], which substantially cause brain damage and cognitive changes [62]. The anti-glycation effect of POL has been documented [18], which also contributes considerably to protecting the brain from the HC+HG posed challenges. Additionally, improved liver health in response to POL is a crucial event that contributes to brain protection. The notion aligns with the literature, which suggests a strong axis between the liver and brain [63]. Fatty liver disease facilitates events that impact the permeability of the blood–brain barrier (BBB), eventually leading to the accumulation of toxic substances and inflammatory cells in the brain [64], which can cause severe damage to the brain.

High-cholesterol and high-galactose have been recognized to have a detrimental impact on the eyes, including retinal damage and cataracts [8,65]. Notably, the ocular section, which includes the retina and cornea, is highly susceptible to oxidative stress under the influence of hypercholesterolemia and diabetes [65], a process that enhances lipid peroxidation and generates 4-hydroxyalkenals, such as 4-HNE [66]. Accumulation of 4-HNE has been described to induce oxidative stress in the cornea and retina, leading to apoptosis through the induction of the caspase-3-mediated pathway [67]. Herein, the consumption of HC+HG resulted in severe ROS generation, apoptosis, and accumulation of 4-HNE in the retina, accompanied by notable corneal opacity in the ocular section of zebrafish. The consumption of POL showed a promising impact against HC+HG-induced adverse events and effectively prevented corneal opacity. We speculate that POL, owing to its cellular antioxidant properties, mediates a range of events that contribute to ocular protection. This notion aligns with an earlier report documenting the impact of antioxidants on preventing eye damage [65]. One study conducted in diabetic rats demonstrated the curative effect of cacao on ocular lens opacity, mediated by its antioxidant properties and the inhibition of oxidative stress [68]. Despite the impact on ROS generation and antioxidants, POL’s positive effect on augmenting HDL might be a crucial factor in eye-protective activities, as few studies have established a correlation between corneal opacity and HDL-C [69,70]. Nevertheless, to reach a confirmatory conclusion, further detailed mechanistic investigations are warranted. To the best of our knowledge, this is one of the first reports suggesting that POL may help protect against ocular damage induced by a high-cholesterol and high-galactose diet.

Although zebrafish share many physiological and genetic similarities with humans, some important differences must be considered when interpreting translation outcomes. For instance, in zebrafish, apolipoprotein accounts for 36% of plasma proteins, compared to only 10% in humans, and fish LDL contains a high level of triglycerides and a low level of cholesterol ester [71]. Most importantly, zebrafish are cold-blooded (poikilothermic) [72], whereas humans are warm-blooded (homeothermic); therefore, the metabolic rates in zebrafish vary substantially due to the environmental conditions [73], contrary to the relatively stable metabolic rate in humans. Also, adipose tissue development and distribution are considerably different between the two species [72]. Moreover, zebrafish harbor nucleated erythrocytes and thrombocytes, which differ from the enucleated ones in humans [74]. Due to these physiological and anatomical differences, findings from zebrafish models should be interpreted carefully when extrapolated to human responses.

Limitations of this study: The absence of a mechanistic role for POL antioxidant activity, the lack of its gender-specific impact on metabolic stress, and the long-term safety aspects of POL supplementation remain key limitations that should be addressed in future studies. Also, comprehensive clinical trials are warranted to validate the current findings in human subjects.

## 5. Conclusions

The 30-week POL supplementation prevents HC+HG-induced hepatomegaly, hepatic inflammation, fatty liver, dyslipidemia, hyperglycemia, brain damage, vision weakness, and augments the antioxidant status. POL successfully minimizes ROS generation and senescence in the brain and mitigates HC+HG-induced cognitive impairment. POL exerted an inhibitory effect on corneal clouding incidence and reduced oxidative stress, apoptosis, and 4-HNE accumulation in the retina. The study concludes that POL’s cellular antioxidant properties confer various beneficial physiological effects, supporting its use as a nutraceutical to prevent metabolic stress and associated disorders.

## Figures and Tables

**Figure 1 antioxidants-14-01453-f001:**
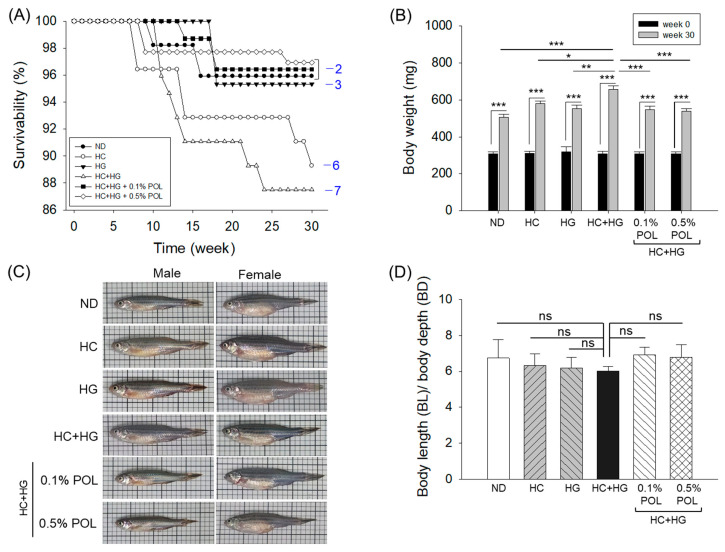
Comparison of zebrafish survivability and body weight across different dietary groups during 30 weeks of feeding. (**A**) Survival kinetics; numerical values in blue font represent the net zebrafish death in the respective groups during the 30 weeks. (**B**) Body weight measurements were taken at the beginning (week 0) and at 30 weeks. (**C**) Exemplary images of male and female zebrafish across different groups post-30 weeks of intake of different diets. (**D**) Morphometry of zebrafish is expressed as the ratio of body length (BL)/body depth (BD). ND represents the normal diet; HC represents high-cholesterol (4%, *w*/*w*) diet; HG represents high-galactose (30%, *w*/*w*) diet; HC+HG represents high-cholesterol (4%, *w*/*w*) mixed with high-galactose (30%, *w*/*w*) diet; and HC+HG+POL (0.1%/0.5%) represent high-cholesterol + galactose diet supplemented with policosanol (0.1% or 0.5% *w*/*w*). The symbols * (*p* < 0.05), ** (*p* < 0.01), and *** (*p* < 0.001) underscore the statistical difference between the groups; “ns” indicates a non-significant difference with respect to the HC+HG group.

**Figure 2 antioxidants-14-01453-f002:**
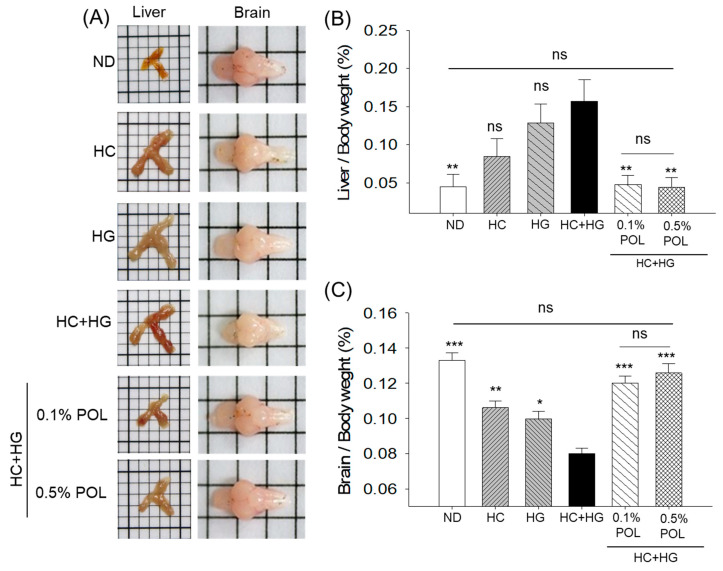
Organ morphology (**A**), and weight of (**B**) liver, (**C**) brain of zebrafish after 30 weeks supplementation of different diets. ND represents the normal diet; HC represents high-cholesterol (4%, *w*/*w*) diet; HG represents high- galactose (30%, *w*/*w*) diet; HC+HG represents high-cholesterol (4%, *w*/*w*) mixed with high-galactose (30%, *w*/*w*) diet; and HC+HG+POL (0.1%/0.5%) represent high-cholesterol + high-galactose diet supplemented with policosanol (0.1% or 0.5% *w*/*w*). The symbols * (*p* < 0.05), ** (*p* < 0.01), and *** (*p* < 0.001) underscore the significant differences relative to the HC+HG group; “ns” highlights non-significant difference.

**Figure 3 antioxidants-14-01453-f003:**
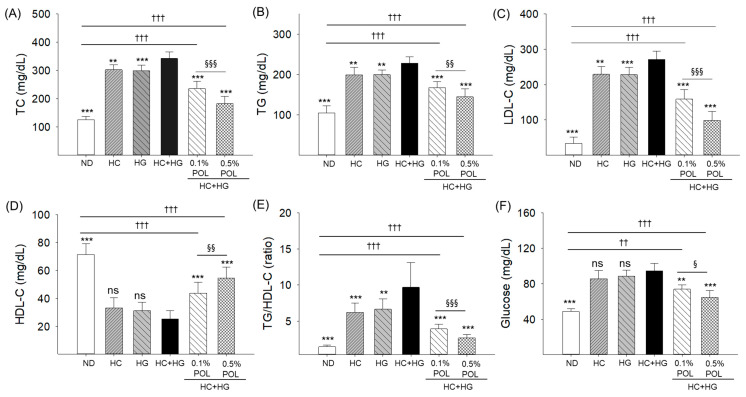
Quantification of zebrafish blood (**A**) total cholesterol (TC), (**B**) triglycerides (TG), (**C**) low-density lipoprotein cholesterol (LDL-C), (**D**) high-density lipoprotein cholesterol (HDL-C), (**E**) ratio of triglycerides with high-density lipoprotein cholesterol (TG/HDL-C), and (**F**) glucose levels after 30 weeks of dietary supplementation. ND represents the normal diet; HC represents high-cholesterol (4% *w*/*w*) diet; HG represents high-galactose (30%, *w*/*w*) diet; HC+HG represents high-cholesterol (4% *w*/*w*) mixed with high-galactose (30%, *w*/*w*) diet, and HC+HG+POL (0.1%/0.5%) represent high-cholesterol + high-galactose diet supplemented with policosanol (0.1% or 0.5% *w*/*w*). The symbols ** (*p* < 0.01), and *** (*p* < 0.001) indicate significant differences relative to the HC+HG group. The symbols ^††^ (*p* < 0.01), and ^†††^ (*p* < 0.001) denote significant differences compared with the ND group. The symbols ^§^ (*p* < 0.05), ^§§^ (*p* < 0.01), and ^§§§^ (*p* < 0.001) represent significant differences relative to the 0.5% POL group; “ns” highlights a non-significant difference.

**Figure 4 antioxidants-14-01453-f004:**
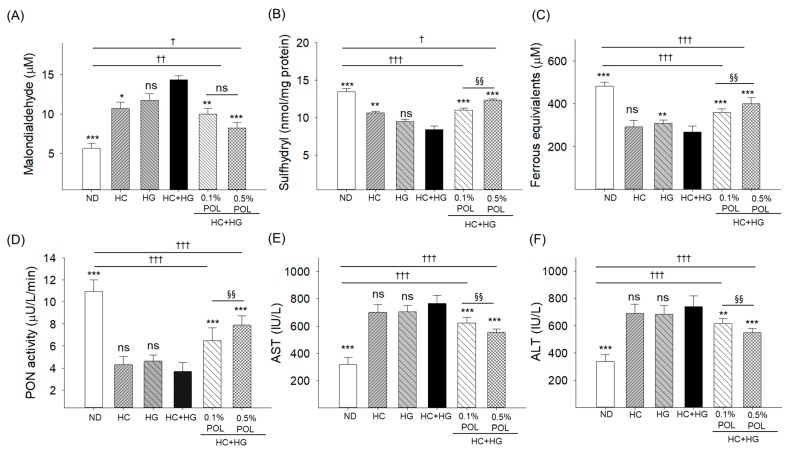
Assessment of zebrafish blood levels of (**A**) malondialdehyde, (**B**) sulfhydryl content, (**C**) ferric ion reduction ability, (**D**) paraoxonase (PON) activity, (**E**) aspartate aminotransferase (AST), and (**F**) alanine aminotransferase (ALT) after 30 weeks of dietary supplementation. ND represents the normal diet; HC represents high-cholesterol (4%, *w*/*w*) diet; HG represents high-galactose (30%, *w*/*w*) diet; HC+HG represents high-cholesterol (4%, *w*/*w*) mixed with high-galactose (30%, *w*/*w*) diet; and HC+HG+POL (0.1%/0.5%) represent high-cholesterol + high-galactose diet supplemented with policosanol (0.1% or 0.5% *w*/*w*). The symbols * (*p* < 0.05), ** (*p* < 0.01), and *** (*p* < 0.001) indicate significant differences relative to the HC+HG group. The symbols ^†^ (*p* < 0.05), ^††^ (*p* < 0.01), and ^†††^ (*p* < 0.001) denote significant differences compared with the ND group. The symbol ^§§^ (*p* < 0.01) represents significant differences relative to the 0.5% POL group; “ns” highlights a non-significant difference.

**Figure 5 antioxidants-14-01453-f005:**
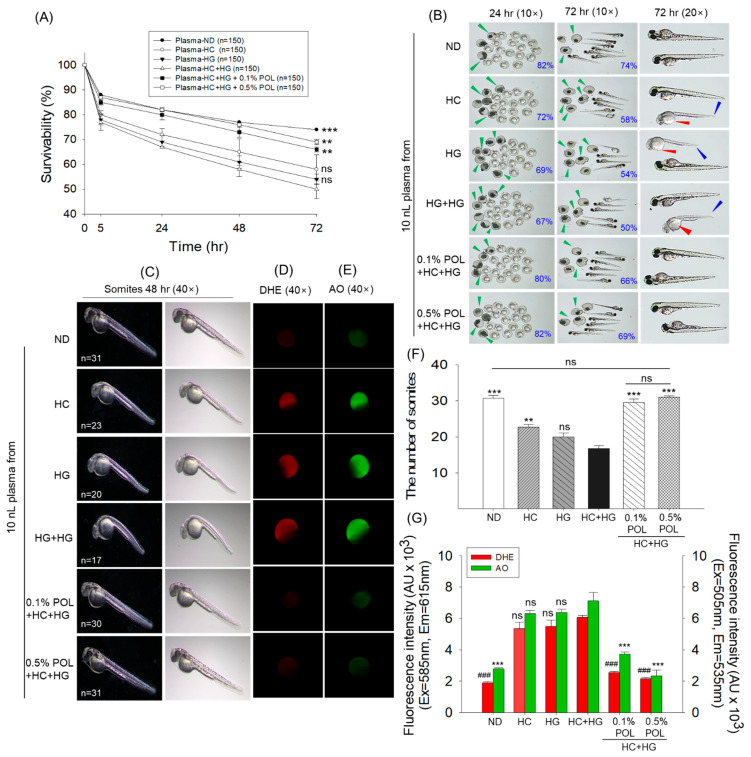
Effect of plasma collected from the zebrafish after 30 weeks of dietary intervention. (**A**) Survival rate of zebrafish embryos (*n* = 150) following microinjection with 10 nL of plasma. (**B**) Exemplary images of the embryo captured at 24 and 72 h post-injections; green arrowhead represents dead embryos while red and blue arrowheads highlight yolk sac edema and tail fin curvature, respectively. (**C**) Somites in the embryos at 48 h post-injection. (**D**,**E**) Dihydroethidium (DHE) and acridine orange (AO) fluorescent staining of embryos. (**F**) Average somite counts in different groups. (**G**) DHE and AO fluorescent intensity quantified by using Image J software (https://imagej.net/ij, version 1.53, assessed on 6 June 2025). ND represents the normal diet; HC represents high-cholesterol (4% *w*/*w*) diet; HG represents high-galactose (30%, *w*/*w*) diet; HC+HG represents high-cholesterol (4% *w*/*w*) mixed with high-galactose (30%, *w*/*w*) diet; and HC+HG+POL (0.1%/0.5%) represent high-cholesterol + high-galactose diet supplemented with policosanol (0.1% or 0.5% *w*/*w*). The symbols ** (*p* < 0.01), and *** (*p* < 0.001) indicate significant differences relative to the HC+HG group. The symbol ^###^ (*p* < 0.001) represents a significant difference relative to the HC+HG group for AO fluorescent intensity; “ns” highlights a non-significant difference.

**Figure 6 antioxidants-14-01453-f006:**
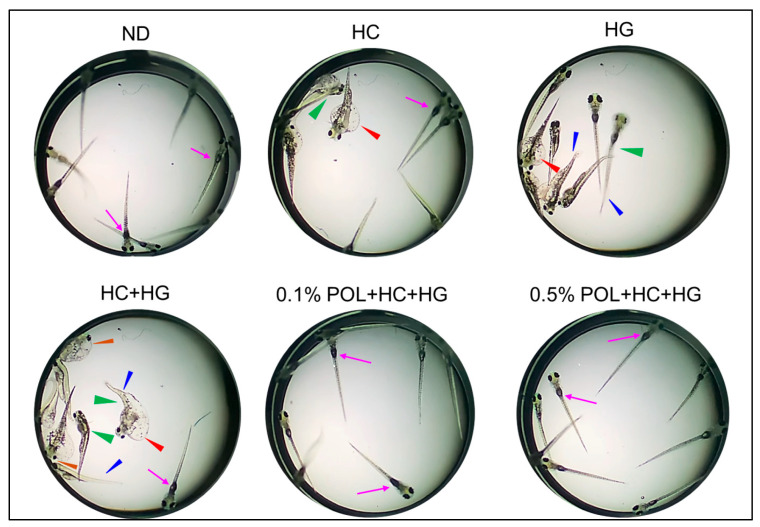
Representative images of embryos at 144 h post-injection with plasma obtained from zebrafish after 30 weeks of intake of the respective diets. The blue arrowhead highlights the tail fin curvature, the green arrowhead depicts the bending of back, the red arrowhead indicates the yolk sac edema, and the blown color arrowhead highlights the pericardial edema. Pink arrows indicate an inflated swimming bladder. ND represents the normal diet; HC represents high-cholesterol (4% *w*/*w*) diet; HG represents high-galactose (30%, *w*/*w*) diet; HC+HG represents high-cholesterol (4% *w*/*w*) mixed with high-galactose (30%, *w*/*w*) diet; and HC+HG+POL (0.1%/0.5%) represent high-cholesterol + high-galactose diet supplemented with policosanol (0.1% or 0.5% *w*/*w*).

**Figure 7 antioxidants-14-01453-f007:**
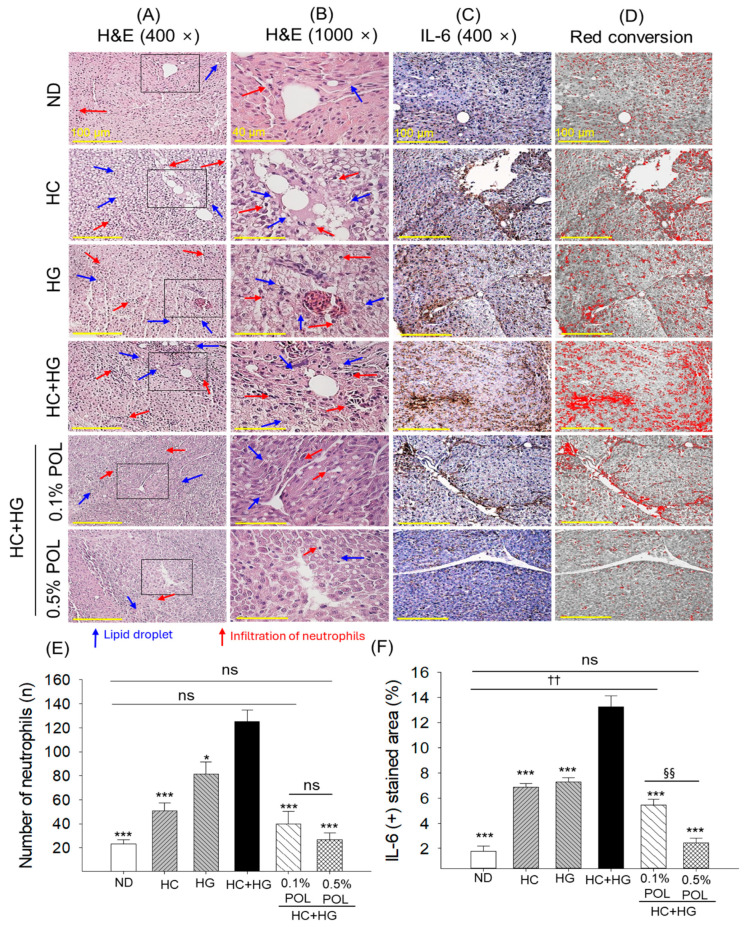
Zebrafish liver histology among the different groups after 30 weeks of intake of specialized diets. (**A**) Hematoxylin and eosin (H&E)-stained liver sections observed at 400× magnification [scale bar, 100 μm]. (**B**) High-magnified view (1000× magnification) of the H&E section covered under the black box (panel A) [scale bar, 40 μm]. (**C**) Immunohistochemical (IHC) images for the detection of interleukin (IL)-6. (**D**) To improve visibility, the IHC-stained area was converted to red (red conversion) at a brown color threshold (20–120) using ImageJ (https://imagej.net/ij, version 1.53, assessed on 6 June 2025). (**E**,**F**) Quantification of neutrophil numbers and IL-6-stained area, respectively. ND represents the normal diet; HC represents high-cholesterol (4% *w*/*w*) diet; HG represents high-galactose (30%, *w*/*w*) diet; HC+HG represents high-cholesterol (4% *w*/*w*) mixed with high-galactose (30%, *w*/*w*) diet; and HC+HG+POL (0.1%/0.5%) represent high-cholesterol + high-galactose diet supplemented with policosanol (0.1% or 0.5% *w*/*w*). The symbols * (*p* < 0.05), and *** (*p* < 0.001) indicate significant differences relative to the HC+HG group. The symbol ^††^ (*p* < 0.01) denotes significant differences compared with the ND group. The symbol ^§§^ (*p* < 0.01) represents significant differences relative to the 0.5% POL group; “ns” highlights a non-significant difference.

**Figure 8 antioxidants-14-01453-f008:**
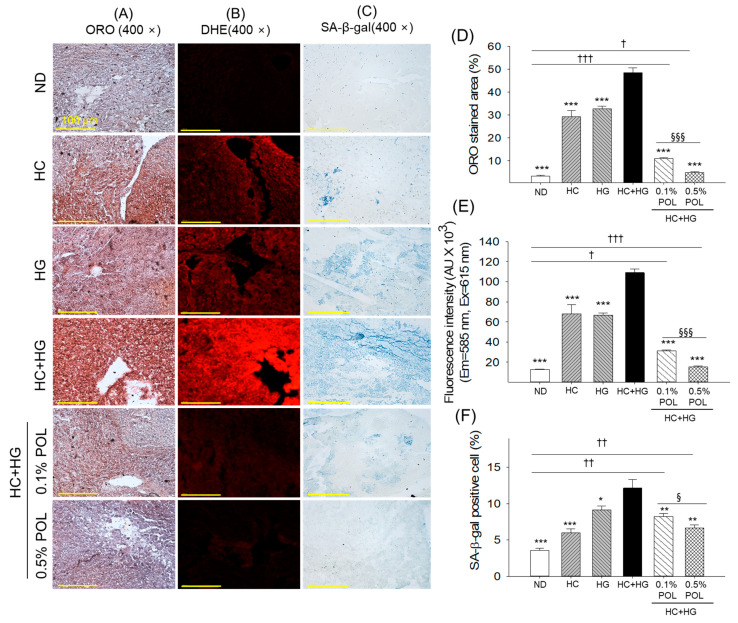
Different staining of the zebrafish liver section after 30 weeks of consuming different diets. (**A**) Oil red O (ORO) staining, (**B**) dihydroethidium (DHE) fluorescent staining, and (**C**) senescent-associated β-galactosidase (SA-β-gal) staining, [scale bar, 100 μm]. Quantification of (**D**) ORO-stained area, (**E**) DHE fluorescent intensity, and (**F**) SA-β-gal positive cells. ND represents the normal diet; HC represents high-cholesterol (4% *w*/*w*) diet; HG represents high-galactose (30%, *w*/*w*) diet; HC+HG represents high-cholesterol (4% *w*/*w*) mixed with high-galactose (30%, *w*/*w*) diet; and HC+HG+POL (0.1%/0.5%) represent high-cholesterol + high-galactose diet supplemented with policosanol (0.1% or 0.5% *w*/*w*). The symbols * (*p* < 0.05), ** (*p* < 0.01), and *** (*p* < 0.001) indicate significant differences relative to the HC+HG group. The symbols ^†^ (*p* < 0.05), ^††^ (*p* < 0.01), and ^†††^ (*p* < 0.001) denote significant differences compared with the ND group. The symbols ^§^ (*p* < 0.05), and ^§§§^ (*p* < 0.001) represent significant differences relative to the 0.5% POL group.

**Figure 9 antioxidants-14-01453-f009:**
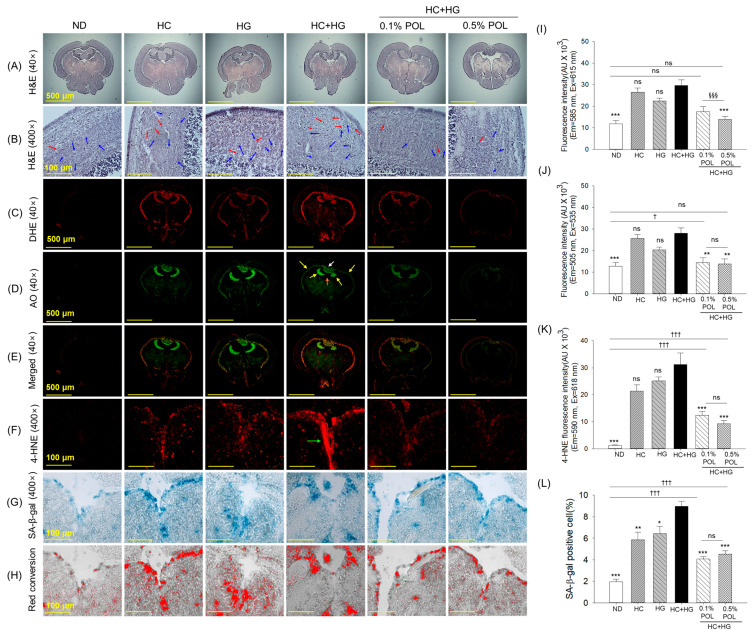
Histology of the zebrafish brain section after 30 weeks of consumption of different diets. (**A**,**B**) display hematoxylin and eosin staining (H&E) observed at 40× and 400× magnification, respectively; the blue and red arrows highlight the vacuolation and mononuclear cells with a clear zone, respectively. (**C**) Dihydroethidium (DHE) and (**D**) acridine orange (AO) fluorescent staining, and (**E**) merged images of DHE- and AO-stained sections. The yellow, orange, and white arrows indicate the periventricular gray zone (PGZ) of tectum opticum (TeO), torus longitudinalis (TL), and vascular cerebelli (Val), respectively. (**F**) Immunohistochemical (IHC) staining for 4-hydroxynonenal (4-HNE); the green arrow highlights the vascular lacuna of the area postrema (Vas). (**G**) Senescent-associated β-galactosidase (SA-β-gal) staining. (**H**) Red conversion of the SA-β-gal-stained area using Image J software (https://imagej.net/ij, version 1.53, assessed on 6 June 2025) at a blue color threshold value of 0–120. Quantification of (**I**) DHE, (**J**) AO, and (**K**) 4-HNE fluorescent intensity. (**L**) Quantification of senescent positive cells. ND represents the normal diet; HC represents high-cholesterol (4%, *w*/*w*) diet; HG represents high-galactose (30%, *w*/*w*) diet; HC+HG represents high-cholesterol (4%, *w*/*w*) mixed with high-galactose (30%, *w*/*w*) diet; and HC+HG+POL (0.1%/0.5%) represent high-cholesterol + high-galactose diet supplemented with policosanol (0.1% or 0.5% *w*/*w*). The symbols * (*p* < 0.05), ** (*p* < 0.01), and *** (*p* < 0.001) indicate significant differences relative to the HC+HG group. The symbols ^†^ (*p* < 0.05) and ^†††^ (*p* < 0.001) denote significant differences compared with the ND group. The symbol ^§§§^ (*p* < 0.001) represents significant differences relative to the 0.5% POL group; “ns” highlights a non-significant difference.

**Figure 10 antioxidants-14-01453-f010:**
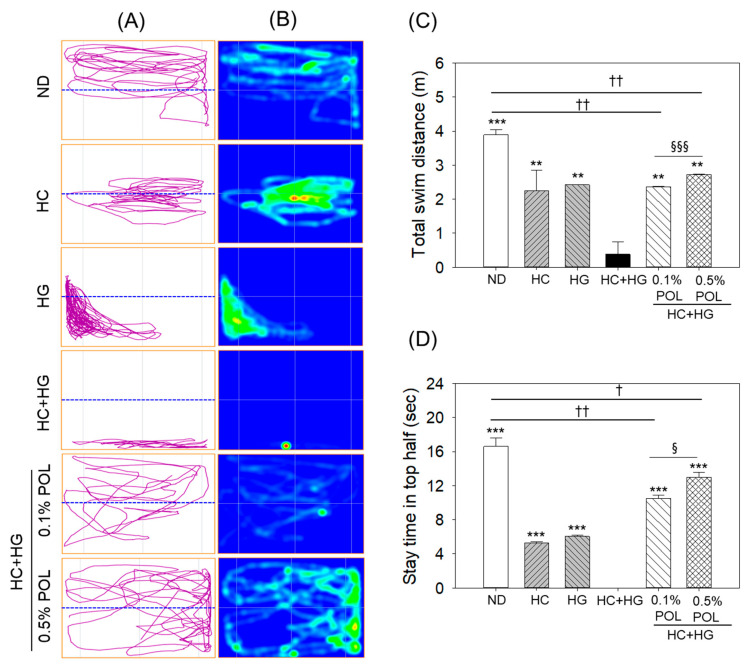
Assessment of zebrafish swimming behavior among different groups consuming the specified diets. (**A**) Representative swimming trajectories of zebrafish using the software (Any-Maze version 7.0, Kim and friends, Seoul, Republic of Korea), the blue dotted line in the image represents the central line dividing the tank into two equal halves (i.e., top and lower). (**B**) Heat map of swimming trajectories. (**C**) Average total swimming distance (m). (**D**) Average time of stay (Sec) of zebrafish in the top half of the tank. The data obtained from the one min of swimming experiment. ND represents the normal diet; HC represents high-cholesterol (4%, *w*/*w*) diet; HG represents high-galactose (30%, *w*/*w*) diet; HC+HG represents high-cholesterol (4%, *w*/*w*) mixed with high-galactose (30%, *w*/*w*) diet; and HC+HG+POL (0.1%/0.5%) represent high-cholesterol + high-galactose diet supplemented with policosanol (0.1% or 0.5% *w*/*w*). The symbols ** (*p* < 0.01) and *** (*p* < 0.001) indicate significant differences relative to the HC+HG group. The symbols ^†^ (*p* < 0.05) and ^††^ (*p* < 0.01) indicate significant differences relative to the ND group. The symbols ^§^ (*p* < 0.05) and ^§§§^ (*p* < 0.001) represent significant differences relative to the 0.5% POL group.

**Figure 11 antioxidants-14-01453-f011:**
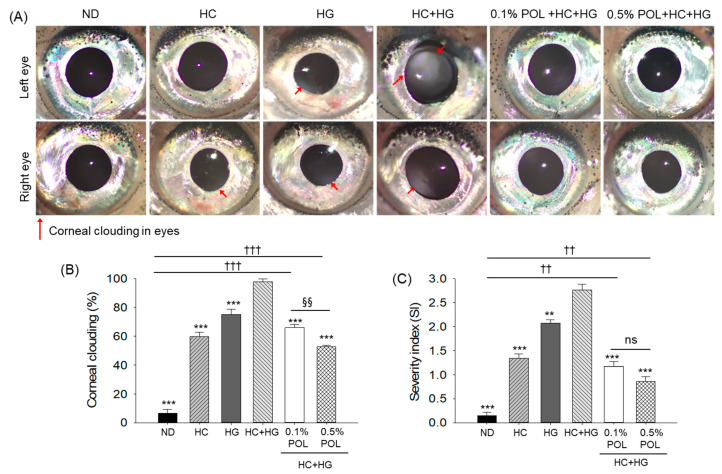
Eye morphology of zebrafish after 30 weeks of consuming the designated diets. (**A**) Representative images of the left and right eyes of zebrafish, with red arrow indicating corneal clouding. (**B**) Quantification of the corneal clouding incidence among the groups. (**C**) Severity index (SI) of clouding in the eyes. ND represents the normal diet; HC represents high-cholesterol (4%, *w*/*w*) diet; HG represents high-galactose (30%, *w*/*w*) diet; HC+HG represents high-cholesterol (4%, *w*/*w*) mixed with high-galactose (30%, *w*/*w*) diet; and HC+HG+POL (0.1%/0.5%) represent high-cholesterol + high-galactose diet supplemented with policosanol (0.1% or 0.5% *w*/*w*). The symbols ** (*p* < 0.01) and *** (*p* < 0.001) indicate significant differences relative to the HC+HG group. The symbols ^††^ (*p* < 0.01) and ^†††^ (*p* < 0.001) indicate significant differences relative to the ND group. The symbol ^§§^ (*p* < 0.01) represents significant differences relative to the 0.5% POL group; “ns” highlights a non-significant difference.

**Figure 12 antioxidants-14-01453-f012:**
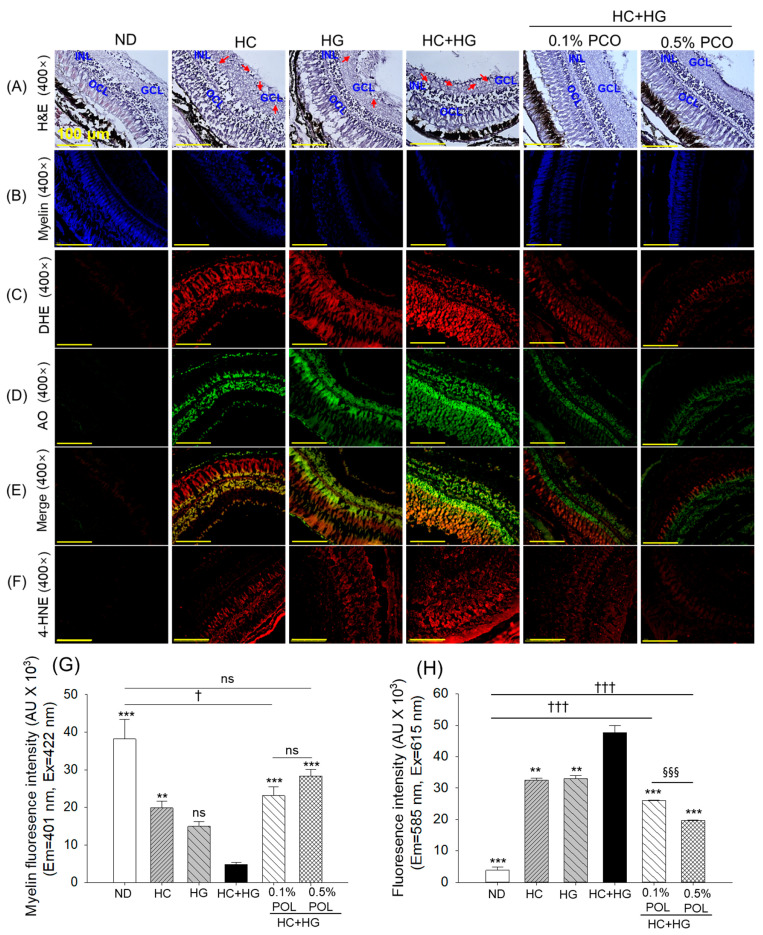
Histological examination of zebrafish eyes following 30 weeks of intake of different diets. (**A**) Hematoxylin and eosin (H&E) staining of the eye (retina); GCL, INL, and ONL are abbreviated for ganglion cell layer, inner nuclear layer, and outer nuclear layer, respectively; the red arrow highlights the vacuolation in the GCL, [scale bar, 100 μm]. (**B**) Immunohistochemical (IHC) images of myelin sheath. (**C**–**E**) Dihydroethidium (DHE), acridine orange (AO) fluorescent intensity, and merged images of DHE and AO staining, respectively. (**F**) Immunohistochemical (IHC) staining to detect 4-hydroxynonenal (4-HNE). Quantification of (**G**) Myelin, (**H**) DHE, (**I**) AO, and (**J**) 4-HNE fluorescent intensities, respectively. ND represents the normal diet; HC represents high-cholesterol (4%, *w*/*w*) diet; HG represents high-galactose (30%, *w*/*w*) diet; HC+HG represents high-cholesterol (4%, *w*/*w*) mixed with high-galactose (30%, *w*/*w*) diet; and HC+HG+POL (0.1%/0.5%) represent high-cholesterol + high-galactose diet supplemented with policosanol (0.1% or 0.5% *w*/*w*). The symbols ** (*p* < 0.01) and *** (*p* < 0.001) indicate significant differences relative to the HC+HG group. The symbols ^†^ (*p* < 0.05), ^††^ (*p* < 0.01), and ^†††^ (*p* < 0.001) represent significant differences relative to the ND group. The symbols ^§§^ (*p* < 0.01) and ^§§§^ (*p* < 0.001) represent significant differences relative to the 0.5% POL group; “ns” highlights a non-significant difference.

## Data Availability

The original contributions presented in this study are included in the article/Appendix A. Further inquiries can be directed to the corresponding author.

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
