# Peer review of "Cuban Sugarcane Wax Alcohol Supplementation Prevents Brain and Eye Damages of Zebrafish Exposed to High-Cholesterol and High-Galactose Diet for 30 Weeks: Protection of Myelin, Cornea, and Retina"

_antioxidants, 2025, doi:10.3390/antiox14121453_

Round 1
Reviewer 1 Report
I recommend the acceptance of this manuscript after minor revisions. The paper presents a solid and well-executed study that makes a valuable contribution to the field.
- The study predominantly attributes the protective effects of POL to its "antioxidant" properties, which appears to be an overarching phenomenon rather than a specific mechanism. How does POL initiate its antioxidant effects? Which key signaling pathways (e.g., Nrf2, AMPK) are involved? There is a lack of molecular biology evidence (e.g., Western Blot, qPCR) to elucidate the underlying mechanisms of action.
- The duration of analysis for the behavioral experiments is not clearly specified in the Methods section, being mentioned only in the Results. This omission affects the reproducibility of the experiments.
- The symbols mentioned in the caption of Figure 3 need to be clearly explained.
- The claim that the ocular protective effects are "the first report" requires more caution. Existing studies have investigated the role of POL or other antioxidants in ocular diseases.
- While the zebrafish model was appropriately selected, the Discussion should more explicitly acknowledge the limitations of the model regarding its physiological differences from humans, and address the considerations necessary for extrapolating the results to humans.
- Future studies of a similar nature should include a POL-alone supplementation group to comprehensively assess its safety profile.
- The figure legends and descriptions require further optimization to ensure clarity for readers.
- Regarding Figure 7: Shouldn't this be a single, consolidated figure? The panels need to be properly aligned. Furthermore, what was the rationale for selecting different scale bars? This requires further justification.
- The clarity of some tissue section images is suboptimal and needs improvement. Consider replacing them with higher-resolution images if possible.
Author Response
Thank you for your insightful comments. Following the reviewer’s suggestion, we made point-to-point response and reflected on revision.
Please find attached doc as our response.

Reviewer 2 Report
General Assessment
This manuscript presents a comprehensive and long-term investigation into the protective effects of Cuban sugarcane–derived policosanol against metabolic, hepatic, neurological, and ocular damage induced by a high-cholesterol and high-galactose diet in zebrafish. The study uses a wide set of analyses, biochemical, histological, immunohistochemical, oxidative stress markers, and behavioral testing, which is a notable strength. The 30-week feeding regimen is unusually long for zebrafish studies and adds robustness to the findings.
The topic fits well within the scope of Antioxidants, especially regarding oxidative stress, lipid metabolism, and neuroprotection. However, important issues concerning statistical rigor, methodological clarity, interpretation of behavioral data, and transparency regarding conflicts of interest must be addressed. Several results also require quantitative backing rather than qualitative description. Overall, the manuscript has potential but requires major revision before it can be considered for publication.
Major Comments
- Conflict of Interest and Transparency
All authors are affiliated with the Raydel HDL Research Institute, and the tested product is Raydel® policosanol.
A clear and explicit Conflict of Interest statement is mandatory.
Additionally:
The manuscript must clarify whether investigators were blinded during histological scoring, IHC analysis, and behavioral experiments.
Funding sources must be specified.
- Justification of the HC+HG Diet Model
The combined 4% cholesterol + 30% galactose diet is extremely high. Authors should:
- Provide literature validation of this model in zebrafish,
- Discuss its physiological relevance to human conditions, and
- Provide evidence that zebrafish indeed absorb and metabolize cholesterol/galactose at these levels.
- Statistical Analysis: Insufficient Detail
The manuscript lacks essential statistical information:
- Exact sample sizes per assay (blood, histology, IHC, behavior) must be reported.
- Specify normality tests used (e.g., Shapiro–Wilk).
- Report Levene’s test results for variance homogeneity.
- Provide effect sizes for ANOVA (η² or partial η²).
- Indicate whether data were analyzed per tank or per individual (to avoid pseudoreplication).
- Mortality-Related Bias
The HC+HG group shows the highest mortality (12.5%).
This may introduce survivorship bias, since only the healthier individuals could survive the 30-week period.
Authors must explicitly discuss this limitation.
- Missing Controls
There is no POL-only control group.
Thus, it is impossible to determine whether policosanol has any baseline metabolic or behavioral effects independent of HC+HG. This is a major design limitation that requires discussion.
- Histology and IHC: Need for Quantitative Analysis
The current histological results are qualitative. For publication in Antioxidants, the following quantifications are necessary:
- Oil Red O: % lipid-positive area.
- Senescence staining: number of β-gal-positive cells per field.
- DHE/AO staining: mean fluorescence intensity.
- Myelin staining: quantified fluorescence or thickness analysis.
- 4-HNE accumulation: fluorescence intensity.
- Quantification should include software name (e.g., ImageJ), threshold method, and blinding.
- Oxidative Stress Markers: Missing Enzymatic Data
The study reports MDA, sulfhydryl content, FRAP/FRA, and PON activity, but omits key antioxidant enzymes (SOD, CAT, GPx).
At minimum, the authors should justify their absence and discuss this limitation.
- Behavioral Data: Misinterpretation
The manuscript claims improvements in "cognitive function" based solely on swimming distance and locomotion patterns.
However, locomotion is not a cognitive metric. It reflects:
- vitality,
- anxiety,
- neuromotor function,
- or general metabolism.
- The term “cognitive impairment” should be replaced with something like “locomotor performance” or “general activity level” unless additional validated cognitive tests are added (e.g., novel object recognition, T-maze).
- Study Design and Experimental Controls
1.1 Lack of a policosanol-only (POL) control
A major limitation of the study is the absence of a group receiving policosanol alone.
Without this group, it is impossible to determine whether policosanol has baseline effects on lipid metabolism, oxidative stress, or behavior independent of HC+HG exposure.
Please discuss this limitation explicitly and clarify whether a POL-only group could be included in future work.
1.2 Diet justification
The use of 4% cholesterol + 30% galactose is unusually high for zebrafish nutrition.
Please provide:
- citations validating this model,
- justification for physiological relevance, and
- discussion of potential confounding effects of extreme dietary stress.
1.3 Randomization, tank effects, and blinding
The Methods section does not describe:
- how fish were allocated to tanks or groups,
- how many tanks were used per condition,
- whether analyses used the individual fish or the tank as the experimental unit,
- whether histology, IHC, and behavioral assessments were performed blinded.
Please include these essential methodological details. They are important to ensure internal validity and avoid pseudoreplication.
1.4 Mortality imbalance
The HC+HG group exhibits higher mortality.
This introduces potential survivorship bias.
Please report mortality statistics clearly and discuss the potential impact on the interpretation of results.
- Methods: Missing Details and Required Additions
2.1 Statistical analysis
Please provide:
- the exact n for each assay (biochemical, oxidative stress, histology, IHC, behavior),
- normality testing method (e.g., Shapiro–Wilk),
- variance homogeneity testing (e.g., Levene’s test),
- post-hoc test details,
- effect sizes for ANOVA results,
- how missing data, outliers, or mortalities were handled.
2.2 Histology and immunofluorescence quantification
Currently, most histological and IHC results are qualitative.
For publication in Antioxidants, please include quantitative analysis, such as:
- % area stained (Oil Red O, DHE, senescence staining),
- fluorescence intensity (4-HNE, myelin staining),
- cell counts when relevant,
- number of fields analyzed per fish,
- software used (e.g., ImageJ) with thresholding methods.
2.3 Environmental conditions
Please provide full details of husbandry:
- pH, conductivity, dissolved oxygen, ammonia/nitrite/nitrate levels,
- water change frequency,
- photoperiod consistency.
These parameters strongly influence zebrafish physiology.
2.4 Diet preparation
Please clarify:
- how the diet was mixed, pelletized, or dried,
- how stability of cholesterol and galactose was ensured during storage,
- whether food intake was monitored quantitatively.
2.5 Behavioral method details
Please specify:
- lighting conditions,
- tank dimensions used during behavioral recording,
- habituation time before recording,
- whether observers were blinded.
- Results: Clarity, Detail, and Interpretation
3.1 Need for quantitative data
Many conclusions rely on descriptive impressions from images.
Quantitative histological/IHC analysis is necessary to support statements such as:
- “severe damage,”
- “improved morphology,”
- “restored myelination,”
- “reduced oxidative stress.”
3.2 Behavioral interpretation
Swimming distance reflects locomotor performance, not cognition.
The manuscript repeatedly refers to “cognitive deficits” or “cognitive protection,” which is not justified by the assay used.
Please revise these conclusions and describe the findings more accurately as changes in locomotion or activity level.
3.3 Overinterpretation of data
Some statements—particularly regarding neuroprotection and aging—are stronger than the data support.
Please temper these claims unless additional evidence is added.
- Discussion and Contextualization
4.1 Study limitations
Please add a dedicated paragraph clearly acknowledging limitations, including:
- lack of POL-only group,
- absence of histological quantification,
- limited oxidative stress biomarkers,
- behavioral test not measuring cognition,
- mortality imbalance,
- possible tank effects.
4.2 Context with existing literature
While the Discussion includes extensive citations, it would benefit from greater:
- integration of zebrafish-specific metabolic and neurobiological literature,
- acknowledgment of the mixed evidence regarding policosanol efficacy in humans and rodents.
4.3 Mechanistic interpretation
Some mechanistic claims are speculative.
Please ensure all interpretations are directly supported by the results or by cited literature.
- Conflict of Interest and Funding Disclosure
All authors appear to be affiliated with Raydel, the manufacturer of the compound under investigation.
A clear Conflict of Interest statement and Funding Disclosure must be added for transparency, as required by MDPI policies.
Author Response

(The authors gave the same response as above.)

Reviewer 3 Report
This study investigates the protective effects of policosanol (POL), a natural mixture of long-chain alcohols from sugarcane wax, against metabolic stress caused by a high-cholesterol and high-galactose (HC+HG) diet in adult zebrafish. After 30 weeks, the HC+HG diet caused increased mortality, dyslipidemia, oxidative stress, liver damage, cognitive decline, and retinal degeneration. Co-supplementation with POL (0.1% or 0.5%) significantly reduced mortality and improved cholesterol, triglycerides, glucose, and oxidative stress markers, while raising HDL-C and antioxidant activity. POL also alleviated fatty liver, inflammation, brain oxidative damage, and behavioral deficits, and prevented corneal and retinal deterioration. Overall, POL demonstrated strong therapeutic potential in counteracting metabolic and oxidative damage induced by a long-term HC+HG diet.
This study provides a valuable and comprehensive evaluation of policosanol’s protective effects against chronic metabolic stress in zebrafish and contributes relevant data to this area of research. However, the manuscript would be strengthened by a clearer discussion of its limitations, more mechanistic insights, and additional methodological detail to improve transparency and reproducibility. For instance, the microinjection of plasma into zebrafish embryos is a technique not previously described that needs more detailed information. Also, the authors could explore better the limitations of zebrafish as a model for human lipid metabolism and the fact that in the study, high doses were used, how can this be translated to humans?
Did the authors look for sex-specific metabolic responses?
Additionally, the following observations are made to provide a better understanding of the study for readers of this journal.
In the Materials and Methods section:
The Statistical Analysis subsection should be revised to include the p-value or significance level applied in the ANOVA and Tukey post hoc tests. This information is necessary for reproducibility and proper interpretation of the results.”
The authors state that they microinjected plasma into zebrafish embryos. I could not identify any prior peer-reviewed protocols describing microinjection of whole blood plasma in zebrafish, and this is not a commonly established method. Could the authors provide more detailed methodology (plasma preparation, viscosity adjustment, needle specifications, and validation of delivery)? Additional methodological detail is needed to assess feasibility and reproducibility.
Regarding the Behavioral analysis, the authors should mention the number of fish used per group and their age.
The correct description of the “myelin-specific primary antibodies (200× diluted)” should be provided.
In the Results section:
Figure 1. D, all the bars are indicated with Ns (non-significant difference) except for the bar corresponding to the HC+HG treatment, please explain.
The microinjected fish were analysed for swimming behavior at 144 hr post-injection; these methods should be explained in the Materials and Methods section.
The limitations of the study should be more clearly described to help readers understand the constraints of the experimental design and interpretation of the results.
Typos to be corrected:
48-well culture pate (line 207)
Line 308, where is (B) should be (D)
Author Response

(The authors gave the same response as above.)

Round 2
Reviewer 2 Report
The authors have sufficiently improved the manuscript, enhancing both its overall quality and its scientific robustness. The revisions introduced effectively address the previously raised points and strengthen the reliability of the presented results.
After an additional check of the references and stylistic aspects, I found no need for further revisions from my side. The manuscript is now clear, coherent, and appropriately structured.
I consider the work ready for publication. I thank the authors for their efforts in improving the manuscript.
The authors have sufficiently improved the manuscript, enhancing both its overall quality and its scientific robustness. The revisions introduced effectively address the previously raised points and strengthen the reliability of the presented results.
After an additional check of the references and stylistic aspects, I found no need for further revisions from my side. The manuscript is now clear, coherent, and appropriately structured.
I consider the work ready for publication. I thank the authors for their efforts in improving the manuscript.
Author Response
Your review comments were helpful to improve our paper.
We truly acknowledge your appreciation and recommendation for the article the publication. Thank you!
Reviewer 3 Report
The authors have provided responses to the majority of the concerns and comments previously outlined.
There are just two additional points that I would like to address.
The authors state that they microinjected plasma into zebrafish embryos and have referenced a previous study from their own group. However, it would be preferable for them to also cite additional studies by other researchers who have used similar techniques involving microinjection of plasma from donor zebrafish into recipient embryos.
In lines 202 and 220 of the Materials and Methods section, the authors mention the use of a “3% w/v sea salt solution.” This concentration is roughly equivalent to full-strength seawater (≈30 g/L), which is substantially higher than the dilute media typically used for zebrafish embryo maintenance (e.g., egg water or E3 medium). Such high salinity is not standard for zebrafish embryos and may adversely affect their development. The authors should clarify or justify the use of this concentration.
Author Response

(The authors gave the same response as above.)
